# Fever, Tachypnea, and Monocyte Distribution Width Predicts Length of Stay for Patients with COVID-19: A Pioneer Study

**DOI:** 10.3390/jpm12030449

**Published:** 2022-03-12

**Authors:** Sheng-Feng Lin, Hui-An Lin, Han-Chuan Chuang, Hung-Wei Tsai, Ning Kuo, Shao-Chun Chen, Sen-Kuang Hou

**Affiliations:** 1Department of Public Health, School of Medicine, College of Medicine, Taipei Medical University, Taipei 110, Taiwan; linshengfeng@tmu.edu.tw; 2School of Public Health, College of Public Health, Taipei Medical University, Taipei 110, Taiwan; 3Department of Emergency Medicine, Taipei Medical University Hospital, Taipei 110, Taiwan; sevenoking219@tmu.edu.tw (H.-A.L.); kevin575168@gmail.com (H.-W.T.); s82025@gmail.com (N.K.); shaochun36@gmail.com (S.-C.C.); 4Graduate Institute of Injury Prevention and Control, College of Public Health, Taipei Medical University, Taipei 110, Taiwan; 5Division of Infectious Diseases, Department of Internal Medicine, Taipei Medical University Hospital, Taipei 110, Taiwan; hanchuan66@gmail.com; 6Department of Emergency Medicine, School of Medicine, College of Medicine, Taipei Medical University, Taipei 110, Taiwan

**Keywords:** coronavirus 2019 (COVID-19), fever, length of stay (LOS), monocyte distribution width (MDW), tachypnea

## Abstract

(1) Background: Our study investigated whether monocyte distribution width (MDW) could be used in emergency department (ED) settings as a predictor of prolonged length of stay (LOS) for patients with COVID-19. (2) Methods: A retrospective cohort study was conducted; patients presenting to the ED of an academic hospital with confirmed COVID-19 were enrolled. Multivariable logistic regression models were used to obtain the odds ratios (ORs) for predictors of an LOS of >14 days. A validation study for the association between MDW and cycle of threshold (Ct) value was performed. (3) Results: Fever > 38 °C (OR: 2.82, 95% CI, 1.13–7.02, *p* = 0.0259), tachypnea (OR: 4.76, 95% CI, 1.67–13.55, *p* = 0.0034), and MDW ≥ 21 (OR: 5.67, 95% CI, 1.19–27.10, *p* = 0.0269) were robust significant predictors of an LOS of >14 days. We developed a new scoring system in which patients were assigned 1 point for fever > 38 °C, 2 points for tachypnea > 20 breath/min, and 3 points for MDW ≥ 21. The optimal cutoff was a score of ≥2. MDW was negatively associated with Ct value (β: −0.32 per day, standard error = 0.12, *p* = 0.0099). (4) Conclusions: Elevated MDW was associated with a prolonged LOS.

## 1. Introduction

Coronavirus disease 2019 (COVID-19), which was first reported in late 2019, rapidly spread worldwide and became a global health threat in early 2020 [1,2,3,4,5]. Toward the beginning of the pandemic, quarantine and supportive care strategies were used to control the spread of COVID-19. Providing medical care to patients with COVID-19 requires additional space for isolation, protective equipment for medical staff, and strict sanitizing procedures. In this context, ward occupancy has become a critical concern. Furthermore, patients with confirmed COVID-19 who require intensive care unit (ICU) admission, especially those with acute respiratory distress syndrome (ARDS) [6], have a longer average length of stay (LOS) and higher COVID-19-related mortality rates [7,8,9]. In this context, the LOSs of patients with COVID-19 has become a key concern [10,11]. 

Studies have indicated that the mean LOS of patients hospitalized for COVID-19 ranges from 5 to 17 days [11,12,13,14,15,16,17,18]. A patient’s LOS is influenced by various clinical and personal factors, and some clinical tools have been developed to predict patients’ LOSs [19,20]. Patients with multiple comorbidities [11,17,18,21,22] and older age [18,23,24] are more vulnerable and associated with prolonged LOS for COVID-19. Moreover, a recent artificial intelligence-based model [25] found respiratory failure with ventilator support was the most critical variable to influence the LOS (45.4 vs. 7.4 days) and the mortality. Older age, impaired kidney function, and higher body mass index are linked to more severe disease [25]. However, there are limited studies on appropriately predicting LOS by using data collected in emergency department (ED) settings [12].

In addition, numerous biomarkers are considered potential predictors of the severity and LOS for patients with COVID-19. Patients with a high severity of COVID-19 presented with higher proportions of increased white blood cell (WBC) counts (11.4% vs. 4.8%) compared to mild to moderate severity [26]. A study [27] showed C-reactive protein (CRP) was useful for monitoring the progression of infection and to detect severe cases of COVID-19 in the early phase [28]. Procalcitonin (PCT) can reflect disease severity and co-infection in patients with COVID-19 [27,29]. Other inflammatory markers, including neutrophil-to-lymphocyte ratio (NLR) [30,31] and platelet-to-lymphocyte ratio (PLR) [31,32], were associated with the severity of COVID-19.

Monocytes, acting as proinflammatory cytokines, increase markedly during early inflammation [33,34] and are the key members of the mononuclear phagocyte system, an essential part of the innate immunity [35,36]. Monocyte distribution width (MDW) has been employed as an early biomarker to evaluate early septic change in ED settings [37,38,39,40]. MDW is responsive to bacterial and viral pathogens [41,42,43], and the magnitude of increase of MDW is correlated with the intensity of infection and sepsis [37,38,39,40]. MDW is strongly associated with COVID-19 [26,44,45] and can be used to distinguish COVID-19 from other upper airway infections [46]. In addition, MDW is found to be positively associated with inflammatory acute phase proteins, including CRP, ferritin, and fibrinogens [44,47]. A study [47] determined MDW > 24 was associated with an unfavorable outcome in COVID-19. To date, limited information regarding how to predict the LOS of patients with COVID-19 has been made available. Our study aimed to investigate whether MDW could be used in ED settings to predict prolonged LOSs among patients with COVID-19.

## 2. Materials and Methods

### 2.1. Design and Participants

From May 2021 through August 2021, patients presenting to our ED with confirmed COVID-19 were enrolled. In our hospital, real-time polymerase chain reaction (PCR) tests were used to confirm each patient’s COVID-19 infection. SARS-CoV-2 RNA was detected with a real-time PCR machine. The Aptima SARS-CoV-2 Assay Kit (Hologic^®^ Panther System) was used. The inclusivity of the Aptima SARS-CoV-2 assay was assessed using in silico analysis of the assay target capture oligos, amplification primers, and detection probes in relation to 49,741 SARS-CoV-2 sequences available in the National Center for Biotechnology Information (NCBI) and Global Initiative on Sharing Avian Influenza Data (GISAID) gene databases as of July 16th, 2020. The automated nucleic acid isolation system and nucleic acid extraction kit (MagPurix 12S System, Taiwan with Zinexts MagPurix Viral/Pathogen Nucleic Acids Extraction Kit B, New Taipei City, Taiwan) was used. A patient tested positive for COVID-19 was defined as cycle of threshold (Ct) < 40. This investigation was a retrospective cohort study that analyzed medical records from the ED of Taipei Medical University Hospital (data were reviewed by authors H-A Lin, H-W Tsai, and S-K Hou). Our hospital is an academic hospital in Taipei, Taiwan, affiliated with Taipei Medical University, and a major tertiary referral hospital with 750 beds.

### 2.2. Data Collection

Data on the patients’ demographic characteristics; physical findings on arrival to the ED; systemic inflammatory response syndrome (SIRS) scores; quick Sequential Organ Failure Assessment (qSOFA) scores; chest X-ray findings; medical comorbidities; the Charlson Comorbidity Index; inflammatory markers, including CRP, PCT, NLR, PLR, red distribution width (RDW), and MDW; and clinical course were collected. All the patients were followed up from admission to the ED to hospital discharge. The length of stay (LOS) was calculated as the overall LOS in hospital. In our hospital, patients with COVID-19 must be admitted to an isolation ward or the ICU within 6 h on arrival to the ED. This study was approved by the Joint Institutional Review Board of Taipei Medical University (reference number: N201904066).

### 2.3. Clinical Spectrum of COVID-19 Infection

According to the COVID-19 treatment guidelines [48] by the United States National Institutes of Health, four classes of clinical spectrum of COVID-19 were identified: mild, moderate, severe, and critical illness (Appendix A). Patients with COVID-19 are allowed to be discharged when they fulfill the two criteria: (1) acute medical condition that was treated and/or did not need attentive care; (2) if they receive two consecutive negative results in PCR tests performed with an interval of 24–48 h.

### 2.4. New Biomarker Measurement

The new biomarker, MDW, has been routinely measured simultaneously with complete blood cell count (CBC) and differential count (DC) in our hospital since January 2020. MDW, CBC, and DC were analyzed using a hematology analyzer (Beckman Coulter UniCel DxH 900, Beckman Coulter Taiwan, Taipei, Taiwan). MDW was estimated as the standard deviation of a set of monocyte cell volumes [37,38,39]. RDW was measured as the coefficient of variation of red blood cell volume. In addition, NLR was estimated by dividing the neutrophil count by the lymphocyte count, and PLR was estimated by dividing the platelet count by the lymphocyte count.

### 2.5. Statistical Analysis

The normality of continuous variables was assessed using the Shapiro–Wilk tests (Appendix A). For the groups of patients with LOSs of >14 days and ≤14 days, continuous variables were analyzed using the Student’s t test if variables fulfilled normal distribution and the Mann–Whitney U test if variables violated normal distribution. Continuous variables were presented as median (interquartile range, IQR). The categorical variables were examined using Pearson’s chi-squared test (or Fisher’s exact test). Univariable and multivariable logistic regression models were used to obtain the odds ratios (ORs) and corresponding 95% confidence intervals (CIs) for predicting an LOS of >14 days. Youden’s index was used to determine the optimal cutoff values (the point with the maximum value of sensitivity + specificity–1). To select appropriate predictors or variables in the multivariable logistic regression models, we adopted the strategies: (1) including all significant predictors (*p* < 0.05) from the univariate analysis in the multivariable logistic regression (Model 1), and (2) applying backward elimination (Model 2). To construct our newly developed scoring system, our strategy was to assign point values to different potential predictors according to their ORs in Model 2 (Model 3). The integrated discriminatory improvement (IDI) test was used to compare the predictive ability between the two models. The diagnostic performance of each model was obtained by calculating areas under the curves (AUCs) of the receiver of operating characteristics (ROC) curves, and the goodness of fit was evaluated using the Hosmer–Lemeshow test. The subgroup analysis for patients transferred to ICU and non-ICU categories was performed. The decision curve analysis was conducted to assess the clinical utility of the logistic regression models by quantifying their net benefits across different threshold probabilities (a probability of an LOS of >14 days vs. ≤14 days). A *p* value < 0.05 was considered statistically significant, and all statistical analyses were performed using SAS 9.4 (Cary, NC, USA).

### 2.6. Validation Study

Three categories of validation study were performed. First, we conducted the decision curve analysis. Second, to verify the relationship between MDW and COVID-19, generalized linear models (GLMs) were used to examine the associations between patients’ MDWs and LOSs and between their MDWs and Ct values from their RT-PCR tests. Third, we tested the newly developed model in an independent set of patients. From September 2021 to January 2022, 37 patients presenting to our ED with confirmed COVID-19 were enrolled and used as an independent set of patients for validation.

## 3. Results

### 3.1. Patient Characteristics

From May 2021 to August 2021, 120 patients with real-time PCR-confirmed COVID-19 visited our ED and were admitted to our hospital (Table 1). The flow diagram of study was shown in Appendix A. Among these patients, the mean and median LOSs were 17.8 ± 13.8 and 12.0 days, respectively. The distribution of LOS was shown in Appendix A. Compared with the group of patients with LOSs of ≤14 days, the group of patients with LOSs of >14 days had an older mean age, higher proportions of female patients, and patients who experienced symptoms of fever at home, dyspnea, and hypertension; a higher average body temperature and desaturated oxygenation level; and higher grades of severity. In the clinical course, the overall morality was 1.3% (15/120), and the group of patients with LOSs >14 days had a higher proportion of patients treated in the ICU and a higher mortality rate than did the group with LOSs ≤14 days. Otherwise, no significant differences in other characteristics were identified between the two groups (Appendix A).

### 3.2. Inflammatory Markers

Regarding the inflammatory markers measured on arrival to the ED, the group of patients with LOSs of >14 days group had a higher average CRP level, NLR, and MDW. The mean WBC, PCT, and PLR were not significantly different between the groups (Table 1).

### 3.3. Predictors of an LOS of >14 Days

Table 2 presents the univariable logistic regression models used to analyze the predictive value of general characteristics and inflammatory markers. Age; body temperature; respiratory rate; oxygen saturation; qSOFA score; Charlson Comorbidity Index; and the inflammatory markers of CRP, NLR, and MDW were significant predictors of an LOS of >14 days. According to Youden’s index, age > 60 years, tachypnea > 20 breaths/min, SpO_2_ < 96%, CRP > 3 mg/dL, NLR > 3, and MDW ≥ 21 remained as significant predictors. While the clinical predictors exhibited a lower sensitivity of 35%–48% and a higher specificity of 70%–90%, the inflammatory markers demonstrated a higher sensitivity of 60%–90% and a lower specificity of 18%–60%. The positive predictive value, negative predictive value, and accuracy of each variable were shown in Appendix A.

### 3.4. Multivariable Models for Predicting an LOS of >14 Days

We analyzed three models for predicting the LOSs of patients with COVID-19 (Table 3). In Model 1, only tachypnea > 20 breath/min remained as a significant predictor. In Model 2, fever >38 °C, tachypnea > 20 breath/min, and MDW ≥2 1 remained as significant predictors after backward elimination was applied. The Models 1 with and without adjustment of the Charlson Comorbidity Index were presented in Appendix A, and both the models showed no significant difference (IDI: –0.06%, *p* = 0.5162). Model 1 and Model 2 exhibited moderate to high diagnostic accuracy.

For Model 3, we developed a new scoring system in which patients were assigned 1 point for fever > 38 °C, 2 points for tachypnea > 20 breaths/min, and 3 points for MDW ≥ 21; the model exhibited moderate to high diagnostic accuracy.

All the models exhibited adequate goodness of fit, and Model 3 had the lowest Akaike information criterion value. As indicated in Table 4, the optimal cutoff for the new scoring system was a score of ≥ 4 for Model 3 (Youden’s index of 36.1%). The ROC curves of Models 2 and 3 are presented in Figure 1.

### 3.5. Subgroup Analysis for the Non-ICU and ICU Categories

We tested Model 3 for patients in the non-ICU and ICU categories (Appendix A). For non-ICU category, Model 3 was effective in predicting LOS of >14 days in terms of continuous scale (OR: 1.78 per score increase, 95% CI, 1.12–2.83, *p* = 0.0142). For ICU category, Model 3 was not effective in predicting LOSs of >14 days. This study had 120 patients and out of them, 85 were in non-ICU and 35 were in ICU categories. However, the sample size in the subgroup analysis was inadequate. The minimum sample size was 116 to attain an α error of 0.05 and a power (1-β) of 0.80 with the OR of 2.10.

### 3.6. Validation Study

The decision curve analysis (Figure 2) reflected the application of the predictors of an LOS of >14 days that were identified using Models 2–4. Fever >38 °C and tachypnea > 20 breath/min had similar predictive values. MDW ≥ 21 exhibited greater predictive value than did fever >38 °C and tachypnea > 20 breaths/min for a probability of prolonged LOS of up to 50%. In the GLM models (Figure 3), MDW was positively associated with LOS (β: 0.70 per day, standard error = 0.28, *p* = 0.0121), and negatively associated with Ct value (β: −0.32 per day, standard error = 0.12, *p* = 0.0099). In addition, we tested Model 3 in a small independent set of 37 patients (Appendix A). Model 3 showed borderline statistical significance for LOS > 14 days (OR: 1.54 per score increase, 95% CI, 0.97–2.45, *p* = 0.0686).

## 4. Discussion

Our scoring system used elevated MDW, tachypnea, and fever as key predictors of prolonged LOS associated with COVID-19. In Model 3, a score of ≥4 was associated an LOS of >14 days. In our hospital, the discharge policy for COVID-19 is in line with most guidelines [49,50]. Studies have reported mean LOSs of patients with COVID-19 of 5–17 days [11,12,13,14,15,16,17,18]; the differences in the reported values can be attributed to the times at which the studies were conducted (e.g., early 2020 vs. late 2021); severity, treatment guidelines, admission, and discharge criteria; regional prevalence of COVID-19; and even regional rates of COVID-19-related mortality [11,12,13,14,15,16,17,18]. More comprehensive information regarding the prediction of the LOSs of patients with COVID-19 can improve the efficiency of medical resource allocation and ameliorate prognosis [51,52].

Temperature and respiratory rate are key signs in the screening and monitoring of inpatients. Fever and respiratory symptoms are the two most common presentations of patients with COVID-19 [53,54,55], and patients with such symptoms also exhibit an additional risk of sepsis change and respiratory failure [56,57]. Although some studies [58,59] indicated respiratory rate was not precisely measured in most conditions, we considered a cutoff of respiratory rate of >22 should be easily assessed by the well-trained nurses and physicians. Other risk scoring tools for assessment of the degree of illness of patient, including the National Early Warning Score (NEWS) [60] and the Modified Early Warning Score (MEWS) [60,61], include the respiratory rate as an essential component.

MDW, a new biomarker used to predict sepsis [40], was used to distinguish COVID-19 from other common respiratory tract infections in a prior study [46]. A higher MDW reflects more active inflammation activity and greater infection severity and organ dysfunction [38,62], factors which contribute to a prolonged LOS. On the other hand, a study [20] proposed PCT could be used to predict the LOSs of patients with COVID-19; however, in our model, PCT was not associated with LOS. Theoretically, PCT can be used as a biomarker of bacterial infections but not of viral infections, and studies have reported that <5% patients with COVID-19 initially present with coinfections of bacterial pneumonia [63,64]. Moreover, RDW was an ineffective biomarker for LOS. The varied sizes of RBC represent the abnormal red blood cell formation and destruction [65], and this was not immediately linked to acute inflammatory status of the septic patients.

In our study, we have proposed the newly developed Model 3 for predicting LOSs of COVID-19 patients. We considered that Model 3 should be more favored in clinical practice due to its simplicity. Lastly, we confirmed our models with three types of validation study: (1) a decision curve analysis; (2) a linear association between MDW and LOS in patients with COVID-19; and (3) a test in an independent set of 37 patients (Appendix A).

Our analysis was also compatible with a study [66] which used the qSOFA to assess the critically ill patients with COVID-19. In this study [66], the mean qSOFA score for patients with and without mechanical ventilation were both < 1 point [66]. All patients with qSOFA scored for respiratory rate ≥ 22/min. Our study reproduced the results (Table 1). Under this circumference, we did not include the qSOFA score due to the high collinearity between qSOFA and respiratory rate.

To our best knowledge, this is the first study to use MDW as a biomarker to predict the LOSs of patients with COVID-19. However, our study has some limitations. First, SARS-CoV-2 mutates rapidly, and the clinical presentations of COVID-19 may, therefore, change over time. The models presented in our study cannot account for viruses with new genotypes that may appear in the future. Second, a patient’s LOS is determined by many factors that may vary by country and time. Our patients completed the second swab within 24–48 h. These patients may have LOS bias of one day due to the measurement time of the second swab. According to the coronavirus disease 2019 (COVID-19) situation report of the World Health Organization (WHO) [67] and the critical care recommendation of the COVID-19 epidemic in China [1], it is the current consensus that ensures patients who discharged with two negative PCR samples at least 24 h apart. In Asia (including China and Taiwan), the discharge standard of two negative tests is still applicable for COVID-19 patients to date in year 2022. Third, our sample size was just enough to justify the subgroup analysis for patients in the non-ICU and ICU categories. A larger sample size will be needed for further study. Nevertheless, MDW exhibits potential predictive value for the LOS of patients hospitalized for COVID-19.

## 5. Conclusions

Elevated MDW was associated with a longer LOS. We developed a new scoring system accounting for MDW, tachypnea, and fever (assigned each 1 point) in which a score of ≥2 points can be used to predict a prolonged LOS (>14 days).

## Figures and Tables

**Figure 1 jpm-12-00449-f001:**
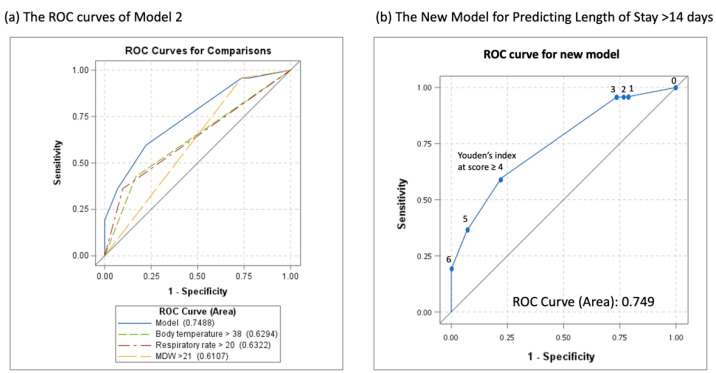
Receiver of operating characteristic curve: (**a**) Model 2; (**b**) the new developed Model 3.

**Figure 2 jpm-12-00449-f002:**
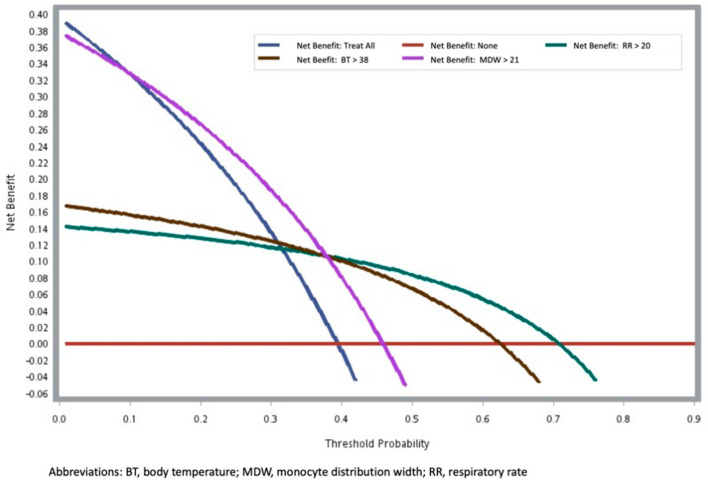
The decision curve analysis.

**Figure 3 jpm-12-00449-f003:**
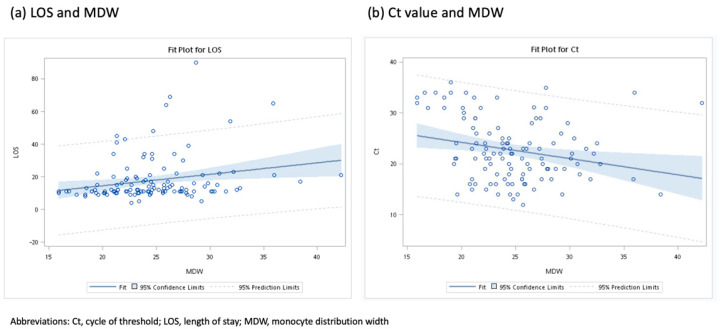
Validation study: (**a**) a positive correlation between length of stay for patients with COVID-19 and MDW; (**b**) a negative correlation between Ct value and MDW.

**Table 1 jpm-12-00449-t001:** Patient characteristics (*n* = 120).

Characteristic	LOS ≤ 14 Days (*n* = 72)	LOS > 14 Days (*n* = 48)	*p* Value
Age (years) †	55.0 (39.0–65.0)	65.0 (54.0–73.0)	0.0036 *
Female sex	40/72 (55.6%)	17/48 (55.6%)	0.0304 *
BMI (kg/m^2^) †	24.5 (21.4–27.8)	24.0 (22.2–28.1)	0.8314
Symptoms			
Fever at home	43/72 (59.4%)	37/48 (77.1%)	0.0481 *
Dyspnea	7/72 (9.7%)	11/48 (22.9%)	0.0107 *
Vital signs at ED			
Body temperature (°C) †	36.8 (36.6–37.5)	37.4 (36.6–38.2)	0.0277 *
Heart rate (beats/min)	86.5 (76.0–100.0)	95.0 (76.0–104.0)	0.3298
Respiratory rate (breaths/min) †	18.0 (17.0–20.0)	20.0 (18.0–24.0)	<0.0001 *
SpO_2_ (%) †	97.0 (95.0–99.0)	96.0 (91.0–98.0)	0.0147 *
SIRS score †	1.0 (1.0–2.0)	2.0 (1.0–2.0)	0.1872
qSOFA score †	0 (0–0)	0 (0–0)	0.0011 *
Glasgow coma scale < 15	2/72 (2.8%)	3/48 (6.3%)	0.2320
Respiratory rate ≥ 22/min	6/72 (8.3%)	17/48 (35.4%)	0.0002 *
SBP ≤ 100 mmHg	2/72 (2.8%)	4/48 (8.3%)	0.1733
Charlson Comorbidity Index †	1.0 (0.0–3.0)	2.5 (1.0–4.0)	0.8724
Severity at ED			<0.0001 *
Mild	39/72 (54.2%)	9/48 (18.8%)	
Moderate	25/72 (34.7%)	11/48 (22.9%)	
Severe	3/72 (4.2%)	10/48 (20.8%)	
Critical	5/72 (6.9%)	18/48 (37.5%)	
Clinical course			
Median LOS (days) †	11.0 (10.0–12.0)	21.0 (17.0–34.0)	<0.0001 *
Transfer to ICU	8/72 (11.1%)	27/48 (56.3%)	<0.0001 *
Mortality	5/72 (6.9%)	10/48 (20.8%)	0.0242 *
Inflammatory markers			
WBC (10^3^ cells/μL) †	6.0 (4.8–7.5)	5.8 (4.7–7.6)	0.8537
RDW (%) †	13.7 ± 1.3	13.5 ± 0.8	0.8724
CRP (mg/dL) †	1.8 (0.3–7.5)	4.9 (1.3–11.4)	0.0051 *
PCT (ng/mL) †	0.06 (0.04–0.11)	0.10 (0.05–0.41)	0.1306
MDW †	23.5 (20.6–26.5)	24.7 (22.3–28.5)	0.0177 *
NLR †	3.5 (1.7–5.7)	4.9 (3.2–9.5)	0.0199 *
PLR †	209.0 ± 159.8	226.6 ± 178.2	0.5095
Median Ct number †	23.0 (18.0–29.0)	21.0 (17.0–24.0)	0.0858

Abbreviations: LOS, length of stay; CCI, Charlson Comorbidity Index; CRP, C-reactive protein; Ct, cycle of threshold value of COVID-19 positive cases; ED, emergency department; ICU, intensive care unit; MDW, monocyte distribution width; NLR, neutrophil-to-lymphocyte ratio; PCT, procalcitonin; PLR, platelet-to-lymphocyte ratio; qSOFA, the quick Sequential Organ Failure Assessment; RDW, red distribution width; SIRS, systemic inflammatory response syndrome; SpO_2_, peripheral capillary oxygen saturation. * Statistical significance (*p* < 0.05). † The Mann–Whitney U test was used.

**Table 2 jpm-12-00449-t002:** Univariable predictors of length of stay >14 days (*n* = 120).

Characteristic	OR (95% CI)	*p* Value	AUC (95% CI)	Cutoff Value ^a^	Sensitivity (95% CI)	Specificity (95% CI)
Age (years)	1.04 (1.01–1.06)	0.0044 *	0.661 (0.561–0.761)	63	-	-
Age > 60 years	3.25 (1.52–6.96)	0.0024 *	0.642 (0.554–0.731)	-	60.4% (45.3–74.2%)	68.1% (56.0–78.6%)
Sex (male vs. female)	2.28 (1.07–4.84)	0.0319 *	0.601(0.511–0.690)	-	-	-
BMI (kg/m^2^)	1.01 (0.93–1.10)	0.7831	0.513 (0.395–0.631)	-	-	-
Body temperature (°C)	1.68 (1.09–2.61)	0.0189 *	0.620 (0.514–0.727)	38	-	-
Fever > 38 °C	3.57 (1.54–8.31)	0.0031 *	0.625 (0.542–0.708)	-	41.7% (27.6–56.8%)	83.3% (72.7–91.1%)
Heart rate (beats/min)	1.01 (0.99–1.03)	0.3273	0.559 (0.451–0.668)	-	-	-
Respiratory rate (breaths/min)	1.33 (1.15–1.55)	0.0002 *	0.726 (0.637–0.815)	20	-	-
Tachypnea > 20 breaths/min	5.09 (1.91–13.55)	0.0011 *	0.629 (0.552–0.705)	-	35.4% (22.2–50.5%)	90.3% (81.0–96.0%)
SpO_2_ (%)	0.87 (0.79–0.96)	0.0043 *	0.633 (0.529–0.737)	96	-	-
SpO_2_ < 96%	2.23 (1.04–4.78)	0.0384 *	0.594 (0.505–0.683)	-	47.9% (33.3–62.8%)	70.8% (58.9–80.5%)
SBP (mmHg)	1.00 (0.99–1.02)	0.8114	0.520 (0.411–0.629)	-	-	-
DBP (mmHg)	0.99 (0.97–1.02)	0.5626	0.540 (0.433–0.646)	-	-	-
MAP (mmHg)	1.00 (0.97–1.02)	0.7877	0.505 (0.397–0.612)	-	-	-
SIRS score (per score)	1.30 (0.87–1.95)	0.2078	0.568 (0.469–0.668)	-	-	-
qSOFA score (per score)	3.82 (1.69–8.63)	0.0012 *	0.634 (0.553–0.715)			
Hypertension	2.43 (1.14–5.20)	0.0222 *	0.604 (0.515–0.693)	-	-	-
Diabetes mellitus	0.96 (0.41–2.29)	0.9299	0.504 (0.426–0.581)	-	-	-
Coronary artery disease	2.33 (0.90–6.07)	0.0825	0.563 (0.490–0.635)	-	-	-
Charlson Comorbidity Index	1.22 (1.01–1.46)	0.0399 *	0.628 (0.529–0.728)	-	-	-
MDW	1.13 (1.04–1.24)	0.0070 *	0.631 (0.531–0.731)	21	-	-
MDW ≥ 21	8.07 (1.78–36.52)	0.0067 *	0.611 (0.552–0.670)	-	95.7% (86.5–99.5%)	27.4% (17.6–39.1%)
WBC	1.00 (1.00–1.00)	0.8872	0.490 (0.383–0.597)	-	-	-
RDW	0.89 (0.64–1.25)	0.5114	0.491 (0.385–0.598)	-	-	-
CRP	1.10 (1.03–1.18)	0.0071 *	0.654 (0.555–0.753)	3	-	-
CRP > 3 mg/dL	2.26 (1.07–4.77)	0.0319 *	0.601 (0.511–0.691)	-	60.4% (45.3–74.2%)	59.7% (47.5–71.1%)
PCT	1.46 (0.87–2.46)	0.1530	0.661 (0.561–0.762)	-	-	-
NLR	1.08 (1.01–1.16)	0.0253 *	0.628 (0.523–0.733)	-	-	-
NLR > 3	3.04 (1.32–7.02)	0.0093 *	0.618 (0.536–0.700)	3	79.2% (65.0–89.5%)	44.4% (32.7–56.6%)
PLR	1.00 (1.00–1.00)	0.5714	0.536 (0.431–0.641)	-	-	-

Abbreviations: AUC; area under curve; CRP, C-reactive protein; DBP, diastolic blood pressure; ED, emergency department; MDW, monocyte distribution width; PCT, procalcitonin; NLR, neutrophil-to-lymphocyte ratio; PLR, platelet-to-lymphocyte ratio; PPV, positive predictive value; NPV, negative predictive value; SBP, systolic blood pressure; SpO_2_, peripheral capillary oxygen saturation; MAP, mean arterial pressure; OR, odds ratio; RDW, red ditribution width. * Statistical significance (*p* < 0.05). ^a^ Cutoff value was determined by Youden’s index.

**Table 3 jpm-12-00449-t003:** Multivariable analysis for predicting length of stay >14 days (*n* = 120).

Characteristic	Model 1 ^a^ OR (95% CI)	*p* Value	Model 2 ^b^ OR (95% CI)	*p* Value	Points Assigned for Model 3 ^c^	Model 3 ^a^ OR (95% CI)
Variable						
Age > 60 years	1.85 (0.55–6.19)	0.3208	-	-	-	-
Sex (male vs. female)	1.48 (0.60–3.67)	0.4001	-	-	-	-
BT > 38 °C	2.46 (0.92–6.55)	0.0717	2.82 (1.13–7.02)	0.0259 *	1	
RR > 20 breaths/min	3.74 (1.12–12.54)	0.0320 *	4.76 (1.67–13.55)	0.0034 *	2	
SpO_2_ < 96%	0.78 (0.28–2.20)	0.6356	-	-	-	-
Hypertension	1.55 (0.60–4.01)	0.3720	-	-	-	-
MDW ≥ 21	4.72 (0.92–24.15)	0.0624	5.67 (1.19–27.10)	0.0296 *	3	
CRP > 3 mg/dL	0.88 (0.29–2.69)	0.8274	-	-	-	-
NLR < 3	1.68 (0.54–5.22)	0.3705	-	-	-	-
Charlson Comorbidity Index	0.95 (0.70–1.30)	0.7545	-	-	-	-
New score (per score)						2.10 (1.48–2.99)
Model fit						
AUC (95% CI)	0.787 (0.701–0.874)		0.749 (0.665–0.833)			0.749 (0.665–0.833)
AIC	161.68		140.97			137.66
Hosmer–Lemeshow test	8.785 (10 groups)	0.3607	3.381 (6 groups)	0.4963		4.270 (6 groups)

Abbreviations: AUC, area under curve; BT, body temperature; CRP, C-reactive protein; NLR, neutrophil-to-lymphocyte ratio; OR, odds ratio; RR, respiratory rate; AIC, Akaike information criterion. ^a^ Model 1 included all significant predictors from the multivariable analysis. ^b^ Model 2 selected variables through backward elimination. Stepwise and forward selection processes selected the same variables. ^c^ Model 3 with new scoring system (point range: 0 to 6). * Statistical significance (*p* < 0.05).

**Table 4 jpm-12-00449-t004:** Diagnostics of new scoring system (Model 3) for predicting length of stay >14 days.

Characteristic	Sensitivity (95% CI)	Specificity (95% CI)	Youden’s Index
Score by using Model 3			
≥1	93.8% (86.9%–100.0%)	22.2% (12.6%–31.8%)	16.0%
≥2	93.8% (86.9%–100.0%)	23.6% (13.8%–33.4%)	17.4%
≥3	93.8% (86.9%–100.0%)	26.4% (16.2%–36.7%)	20.2%
≥4	58.3% (44.4%–72.3%)	77.8% (68.2%–87.4%)	36.1% *
≥5	35.4% (21.9%–49.0%)	93.1% (87.2%–98.3%)	28.4%
≥6	18.8% (7.8%–29.8%)	100.0% (100.0%–100.0%)	18.8%

Abbreviations: AUC, area under curve; CI, confidence interval. * Optimal cutoff of Youden’s index at a score of ≥4 for Model 3 and a score of ≥2.

## Data Availability

Data used in this study were electronic medical chart records and were not available in the public domain. Request of data should formally apply to Office of Human Research, Taipei Medical University, Taipei, Taiwan (tmujirb@gmail.com) and to the Joint Institutional Review Board, Taipei Medical University, Taipei, Taiwan.

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
