# Peer review of "Fever, Tachypnea, and Monocyte Distribution Width Predicts Length of Stay for Patients with COVID-19: A Pioneer Study"

_jpm, 2022, doi:10.3390/jpm12030449_

Round 1

Reviewer 1 Report

The manuscript has been sufficiently improved, thank to the Autors who replied poin to point to the observations.

Author Response

We are grateful for the reviewer's comment.

Reviewer 2 Report

Dear Authors,

I find your work well done. I suggest you read again and correct some little grammar mistakes. The only suggestion I have is to change a single word at line 47 adult respiratory distress syndrome (ARDS),  the correct form is Acute Respiratory Distress Syndrome. Great job!

Author Response

We are grateful for the reviewer's comment. We have corrected the mistake in line 47 to the correct form of acute respiratory distress syndrome (ARDS). Other grammar mistakes were corrected in our revised manuscript as well.

Reviewer 3 Report

The authors have addressed all queries systematically and have made appropriate changes to the revised manuscript. Additional parameters (RDW) were analyzed as suggested, and new scoring models were tested (model 4 for predicting length of stay). In addition, the scoring models have been evaluated on ICU and non-ICU patient data, as suggested; and also tested on newer independent patient sets. The authors have also provided the requested technical details of the SARS-CoV-2 assay and Ct values. Additional references on MDW as a useful biomarker are cited in the revised manuscript as suggested. I recommend accepting the revised manuscript for publication.

Author Response

We are grateful for the reviewer's comment.

This manuscript is a resubmission of an earlier submission. The following is a list of the peer review reports and author responses from that submission.

Round 1

Reviewer 1 Report

Globally, the COVID-19 pandemic has put immense strain on healthcare systems in an unprecedented way. A reliable prediction of the length of stay in the hospital for patients with COVID who present to the hospital is essential to assure adequate bed availability for critically ill patients. In this paper, Sheng-Feng Lin et al, in a retrospective cohort study have systematically analyzed various clinical parameters of 120 COVID-19 positive patients presenting to the emergency department and using statistical test and regression models identified and potential clinical predictors of a length of stay for more than 2 weeks. Among these, the authors identified that the monocyte distribution width (MDW)- a recently identified novel biomarker of sepsis, is a useful predictor of LOS for COVID-19. They also developed an improved scoring system-based multivariable model for predicting an increased LOS using MDW and two other clinical parameters (tachypnea and fever).

MDW has previously been found to be a useful biomarker for severity for COVID-19 and other respiratory infections. Here the authors demonstrate that it could also be potentially used to predict the length of stay of patients. As healthcare systems and personnel are strained by the pandemic, the prediction of LOS is critical for proper patient management and ensuring adequate hospital bed capacities. Overall, the paper is well written, and the analysis of clinical parameters is sound. I recommend this paper for publication, with minor corrections/suggestions for improving the manuscript.

MAJOR COMMENTS

COMMENT 1: The authors have tested three models for predicting LOS, and finally propose model 3 which uses a point-based scoring system for the three parameters-( 1 point for 27 fever >38°C, 2 points for tachypnea >20 breath/min, and 3 points for MDW ≥21). If the authors could comment on the difference in points weightage for the revised scoring system (model 3) for these three parameters, it would be much appreciated. Is it based solely on the odds ratio and the statistical significance of the data of the retrospective study of 120 patients? Are any other factors such as the accuracy of the measurement of these three clinical values taken into consideration?

It would be appreciated if the authors provided more details about the clinical methods and accuracy of the measurement of the respiratory rate of the patients in the study considering the respiratory rate was given a weightage of 2 points in their prediction model. A few recent reports indicate that currently used clinical methods do not accurately measure respiratory rate.

https://doi.org/10.1183/23120541.00023-2020

https://doi.org/10.1111/jan.14584

COMMENT 2Section 2.1 - Lines 86-87: "SARS-CoV-2 RNA was detected with a real-time PCR machine (MagPurix 12S System and Zinexts MagPurix Viral/Pathogen Nucleic Acids Extraction Kit B, Taiwan)" . Here the authors have referred to the name of the automated nucleic acid isolation system and nucleic acid extraction kit used, not the real-time PCR system. If the authors are also able to provide the details of the SARS-CoV-2 detection kit used as well as the SARS-CoV-2 gene/genes in the kit used to confirm the patient's COVID-19 infection, it will be of great help.

COMMENT 3: As the SARS-CoV-2 RT-PCR Ct value varies with different makes of SARS CoV2 RT-PCR assay kits and genes used. It is presumed that the authors have used the same assay for SARS-CoV-2 viral RNA detection throughout the study. I suggest the authors mention the Ct value cut-off used to assign a patient positive in Figure 3b (Ct value versus MDW plot ) and identify the SARS-CoV-2 gene used in the SARS CoV2 RT-PCR assay for which the Ct value was obtained in the figure legend. The average range of Ct values in Table 1 for both groups (LOS ≤ 14 days and LOS> 14 days) would also be helpful.

COMMENT 4: Red Blood Cell Distribution Width (RDW) has also been reported to be associated with disease severity in hospitalized COVID-19 patients and in patients presenting to the hospital emergency departments. It would be helpful if the authors discussed whether incorporating the RDW data into their current scoring system (currently with 3 parameters - MDW, tachypnea, and fever) would improve its ability to predict longer hospital stays. It is assumed that automated hematology analyzers such as Beckman Coulter 119 UniCel DxH 900 systems are equipped with a module for RDW data. In comparison to other parameters like procalcitonin (PCT) and soluble urokinase plasminogen activator receptor (su-PAR), the RDW data would be easy to obtain in most EDs and first-line treatment centers.

COMMENT 5: For COVID-19, the severity of illness and treatment management have also been reported to differ between ICU and non-ICU patients; which in turn influences the length of hospital stay (LOS). As per table 1, roughly 30% (29 of the 120) patients were transferred to the ICU. It would be worthwhile to examine the correlation between the three parameters in model 3 independently / overall score using model 3 among patients in the ICU and non-ICU categories in this study.

COMMENT 6 : Did the authors test their prediction model on an independent set of patient sample data (apart from the 120 patients included in the study)? The prediction score (based on fever, tachypnea, and MDW) made with a smaller independent group of patients can be compared with the actual length of stay at the hospital data, and the results of the model can be independently evaluated.

MINOR COMMENTS

The manuscript could be further improved by citing a few recent research papers that discuss MDW as a useful biomarker for sepsis/clinical severity in COVID-19.

  1. Riva et al, (2021) Scientific Reports 11:12716

https://doi.org/10.1038/s41598-021-92236-6

  1. Alsuwaidi et al. (2022) BMC Infectious Diseases 22:27

https://doi.org/10.1186/s12879-021-07016-4

Author Response

We thank the reviewers for the constructive comments. We have made revisions to the manuscript to address all the questions and comments raised by the reviewers. We have highlighted changes made to the original version by setting the text color to red in the revised manuscript. Our specific responses to each comment are as follows:

Globally, the COVID-19 pandemic has put immense strain on healthcare systems in an unprecedented way. A reliable prediction of the length of stay in the hospital for patients with COVID who present to the hospital is essential to assure adequate bed availability for critically ill patients. In this paper, Sheng-Feng Lin et al, in a retrospective cohort study have systematically analyzed various clinical parameters of 120 COVID-19 positive patients presenting to the emergency department and using statistical test and regression models identified and potential clinical predictors of a length of stay for more than 2 weeks. Among these, the authors identified that the monocyte distribution width (MDW)- a recently identified novel biomarker of sepsis, is a useful predictor of LOS for COVID-19. They also developed an improved scoring system-based multivariable model for predicting an increased LOS using MDW and two other clinical parameters (tachypnea and fever).

MDW has previously been found to be a useful biomarker for severity for COVID-19 and other respiratory infections. Here the authors demonstrate that it could also be potentially used to predict the length of stay of patients. As healthcare systems and personnel are strained by the pandemic, the prediction of LOS is critical for proper patient management and ensuring adequate hospital bed capacities. Overall, the paper is well written, and the analysis of clinical parameters is sound. I recommend this paper for publication, with minor corrections/suggestions for improving the manuscript.

MAJOR COMMENTS

  • COMMENT 1: The authors have tested three models for predicting LOS, and finally propose model 3 which uses a point-based scoring system for the three parameters-( 1 point for fever >38°C, 2 points for tachypnea >20 breath/min, and 3 points for MDW ≥21). If the authors could comment on the difference in points weightage for the revised scoring system (model 3) for these three parameters, it would be much appreciated. Is it based solely on the odds ratio and the statistical significance of the data of the retrospective study of 120 patients? Are any other factors such as the accuracy of the measurement of these three clinical values taken into consideration? It would be appreciated if the authors provided more details about the clinical methods and accuracy of the measurement of the respiratory rate of the patients in the study considering the respiratory rate was given a weightage of 2 points in their prediction model. A few recent reports indicate that currently used clinical methods do not accurately measure respiratory rate.

https://doi.org/10.1183/23120541.00023-2020
https://doi.org/10.1111/jan.14584

  • Thank you for the comment. In our revised manuscript, we have presented two strategies for assigning the scoring points in the newly developed model: (1) assigning point values to different potential predictors according to their ORs in Model 2 (Model 3), and (2) assigning the equal point values to potential predictors (Model 4).
  • In Model 3, we had assigned a higher weight on tachypnea had two reasons. First, tachypnea showed the higher magnitude of OR for LOS. Second, we considered admission respiratory status is highly associated with severity and mortality of COVID-19 patients[1]. Moreover, severe patients with COVID-19 frequently showed volume depletion[2,3], and tachypnea[2], which may exacerbates insensible water loss and hypotension[2,3]. Tachypnea is also the immediate red flag sign for emergency department (ED)[4]. Our model was constructed to predict the LOSs for patients with COVID-19, and we considered to offer a simple guide for solving high bed occupancy rate in the ED. For adults, a normal respiratory rate is between 12 and 20 breaths/min, and a respiratory rate > 20 breath/min means physiological tachypnea. Despite not precisely measured[5,6], a cutoff value respiratory rate > 20 breath/min should not be difficult to be assessed by the well-trained nurses and physicians. In addition, other risk scoring tools for assessment of the degree of illness of patient, including the National Early Warning Score (NEWS)[7,8] and the Modified Early Warning Score (MEWS)[8,9], include the respiratory rate as an essential component.
  • Moreover, we appreciated your comments and tried our best to simplify the model. In Model 4m equal score was assigned to three items of fever > 38°C, tachypnea > 20 breath/min, and MDW ≥21. We have presented the comparison of Models 3 and 4 as follows. First, the diagnostic performance between Models 3 (AUC: 0.749, 95% CI: 0.665-0.883) and Model 4 (AUC:0.736, 95% CI: 0.652-0.821) showed no significant difference (IDI: 0.02%, P = 0.9666). In fact, there was no much difference for the diagnostic performance among score 1-3 in Model 3 (Figure 1). In discussion section, we have proposed the Model 4 being our most favored model for its simplicity. (Please see Table 4, 5, and Figure 1).

Table 4. New scoring system for predicting length of stay >14 days (N = 120).

Characteristic

Model 3a

OR (95% CI)

P value

Model 4b

OR (95% CI)

P value

 New score (per score)

2.10 (1.48–2.99)

<0.0001*

3.86 (2.12–7.03)

<0.0001*

Model fit

 AUC (95% CI)

0.749 (0.665–0.833)

0.736 (0.652–0.821)

 AIC

137.66

137.78

 Hosmer–Lemeshow test

4.270 (6 groups)

2.222 (4 groups)

0.3291

 IDI test

Reference group

-

0.02%

0.9666

Abbreviations: AIC, Akaike information criterion; AUC, area under curve; IDI, integrated discrimination improvement ; OR, odds ratio; RR, respiratory rate.

aModel 3 with new scoring system (point range: 0 to 6).

bModel 4 with new scoring system (point range: 0 to 3).

*Statistical significance (P < 0.05).

Table 5. Diagnostics of new scoring system (Model 3) for predicting length of stay >14 days.

Characteristic

Sensitivity (95% CI)

Specificity (95% CI)

Youden’s index

Score by using Model 3

  ≥1

93.8% (86.9%–100.0%)

22.2% (12.6%–31.8%)

16.0%

  ≥2

93.8% (86.9%–100.0%)

23.6% (13.8%–33.4%)

17.4%

  ≥3

93.8% (86.9%–100.0%)

26.4% (16.2%–36.7%)

20.2%

  ≥4

58.3% (44.4%–72.3%)

77.8% (68.2%–87.4%)

36.1%*

  ≥5

35.4% (21.9%–49.0%)

93.1% (87.2%–98.3%)

28.4%

  ≥6

18.8% (7.8%–29.8%)

100.0% (100.0%–100.0%)

18.8%

Score by using Model 4

  ≥1

93.8% (86.9%–100.0%)

22.2% (12.6%–31.8%)

16.0%

  ≥2

58.3% (44.4%–72.3%)

77.8% (68.2%–87.4%)

36.1%*

  ≥3

18.8% (7.8%–29.8%)

100.0% (100.0%–100.0%)

18.8%

Abbreviation: AUC, area under curve; CI, confidence interval.

*Optimal cutoff of Youden’s index at a score of ≥4 for Model 3 and a score of ≥2.

  • Lastly, we have provided the detailed data regarding our diagnostic accuracy for fever > 38°C, tachypnea > 20 breath/min, and MDW ≥21 in Table 2.

  • COMMENT 2Section 2.1 - Lines 86-87: "SARS-CoV-2 RNA was detected with a real-time PCR machine (MagPurix 12S System and Zinexts MagPurix Viral/Pathogen Nucleic Acids Extraction Kit B, Taiwan)". Here the authors have referred to the name of the automated nucleic acid isolation system and nucleic acid extraction kit used, not the real-time PCR system. If the authors are also able to provide the details of the SARS-CoV-2 detection kit used as well as the SARS-CoV-2 gene/genes in the kit used to confirm the patient's COVID-19 infection, it will be of great help.
  • Thank you for the comment. In our revised manuscript, we have provided information of the SARS-CoV-2 detection kit. The Aptima SARS-CoV-2 Assay Kit (Panther® System by Hologic, Inc) was used. The inclusivity of the Aptima SARS-CoV-2 assay was assessed using in silico analysis of the assay target capture oligos, amplification primers, and detection probes in relation to 49,741 SARS-CoV-2 sequences available in the National Center for Biotechnology In-formation (NCBI) and Global Initiative on Sharing Avian Influenza Data (GISAID) gene databases as of July 16th, 2020. (Please see methods section, 2.1)

COMMENT 3: As the SARS-CoV-2 RT-PCR Ct value varies with different makes of SARS CoV2 RT-PCR assay kits and genes used. It is presumed that the authors have used the same assay for SARS-CoV-2 viral RNA detection throughout the study. I suggest the authors mention the Ct value cut-off used to assign a patient positive in Figure 3b (Ct value versus MDW plot) and identify the SARS-CoV-2 gene used in the SARS CoV2 RT-PCR assay for which the Ct value was obtained in the figure legend. The average range of Ct values in Table 1 for both groups (LOS ≤ 14 days and LOS> 14 days) would also be helpful.

  • Thank you for the comment. In our study, a patient tested positive was defined as Ct value of < 40. For the enrolled 120 patients, the minimum Ct value was 12, and the maximum Ct value was 36 (Supplemental Table I). In our revised manuscript, we have provided and compared the Ct values for both groups. The mean Ct values (23.6 ± 6.4 vs. 21.4 ± 5.2, P = 0.0546) and median Ct values (23 vs. 21, P = 0.0858) showed no significant difference between two groups (Please see Table 1).

COMMENT 4: Red Blood Cell Distribution Width (RDW) has also been reported to be associated with disease severity in hospitalized COVID-19 patients and in patients presenting to the hospital emergency departments. It would be helpful if the authors discussed whether incorporating the RDW data into their current scoring system (currently with 3 parameters - MDW, tachypnea, and fever) would improve its ability to predict longer hospital stays. It is assumed that automated hematology analyzers such as Beckman Coulter 119 UniCel DxH 900 systems are equipped with a module for RDW data. In comparison to other parameters like procalcitonin (PCT) and soluble urokinase plasminogen activator receptor (su-PAR), the RDW data would be easy to obtain in most EDs and first-line treatment centers.

  • Thank you for the comment. The red distribution width (RDW) is a measure of the range of variation of red blood cell volume, and the Beckman Coulter 119 UniCel DxH 900 systems offered the values of coefficient of variation of RDW. In our revised manuscript, we have included RDW into our analysis. Between the groups of LOS ≤ 14 days and LOS > 14 days, the mean RDW values showed no significant difference (13.7 ± 1.3% and 13.5 ± 0.8%, P = 0.8724). In the logistic regression analysis, RDW was not significantly associated with LOS > 14 days (OR: 0.89, 95% CI, 0.64–1.25, P = 0.5114), and area under curve for predicting LOS > 14 days was 0.491 (95% CI, 385–0.598). Generally, a varied size of RBC volume or anisocytosis represents the abnormal red blood cell formation and destruction, and this is not directly linked to inflammatory status of the patients. Consequently, we considered RDW was not a useful marker for predicting LOS in COVID-19 patients. (Please see Table 1, 2 )

COMMENT 5: For COVID-19, the severity of illness and treatment management have also been reported to differ between ICU and non-ICU patients; which in turn influences the length of hospital stay (LOS). As per table 1, roughly 30% (29 of the 120) patients were transferred to the ICU. It would be worthwhile to examine the correlation between the three parameters in model 3 independently / overall score using model 3 among patients in the ICU and non-ICU categories in this study.

  • Thank you for the comment. Of the enrolled 120 patients in our study, 85 patients were treated in the acute medical wards (non-ICU) and 35 patients were in the ICU. In our revised manuscript, we have tested the Models 3 and 4 in the ICU and non-ICU categories. For non-ICU category, both the Models 3 and 4 were effective in predicting LOS of > 14 days in terms of continuous scale (OR: 1.78 per score increase, 95% CI, 1.12-2.83, P = 0.0142), and of the optimal cutoff at Score ≥ 4 in Model 3 and with Score ≥ 2 in Model 4 (OR: 4.05, 95% CI, 1.35-12.12, P = 0.0124). For ICU category, both the Models 3 and 4 were not effective in predicting LOSs of > 14 days. We considered that this difference was caused by most patients in the ICU category had prolonged clinical course. Compared to the group of LOS ≤ 14 days, the group of LOS > 14 days had higher proportions of patients in the ICU (11.1% vs. 56.3%, P < 0.0001). On the other hands, this study had 120 patients and out of them, 85 were in non-ICU and 35 were in ICU categories. However, the sample size in the sub-group analysis was inadequate. The minimum sample size was 116 to attain an α error of 0.05 and a power (1-β) of 0.80 with the OR of 2.10. (Please see Table 1, Supplemental Table III)

Table III. Test of Model 3 in Subgroups of Non-ICU and ICU categories

Characteristic

LOS ≤ 14 days

LOS > 14 days

OR (95% CI)

P value

AUC
(95% CI)

Non-ICU category (N = 85)

Model 3

 Score (per score)

-

-

1.78  (1.12-2.83)

0.0142*

0.693 (0.565-0.812)

   Score ≥ 4

10/64 (15.6%)

9/21 (42.9%)

4.05 (1.35-12.12)

0.0124*

0.636 (0.519-0.754)

Model 4

  Score (per score)

3.09  (1.40-6.84)

0.0054*

0.681 (0.552-0.811)

  Score ≥ 2

10/64 (15.6%)

9/21 (42.9%)

4.05  (1.35-12.12)

0.0124*

0.636 (0.519-0.754)

ICU category (N = 35)

Model 3

   Score (per score)

-

-

1.21(0.59-2.50)

0.6034

0.560 (0.365-0.755)

   Score ≥ 4

6/8 (75.0%)

19/27 (70.4%)

0.79 (0.13-4.79)

0.7993

0.523 (0.340-0.706)

Model 4

  Score (per score)

-

-

1.58(0.49-5.14)

0.4479

0.574 (0.400-0.748)

  Score ≥ 2

6/8 (75.0%)

19/27 (70.4%)

0.79 (0.13-4.79)

0.7993

0.523 (0.340-0.706)

Abbreviations: AUC, area under curve; ICU, intensive care unit; OR, odds ratio.

*Statistical significance (P < 0.05).

COMMENT 6: Did the authors test their prediction model on an independent set of patient sample data (apart from the 120 patients included in the study)? The prediction score (based on fever, tachypnea, and MDW) made with a smaller independent group of patients can be compared with the actual length of stay at the hospital data, and the results of the model can be independently evaluated.

  • Thank you for the comment. From September, 2021 to January, 2022, a total of 37 patients with COVID-19 consecutively visited our emergency department (ED) and were admitted in our hospital. In our revised manuscript, we have analyzed the 37 patients as the validation study as follows. While the Model 3 showed borderline statistical significance for LOS > 14 days (OR: 1.54 per score increase, 95% CI, 0.97-2.45, P = 0.0686) the Model 4 exhibited statistically significance for LOS > 14 days (OR: 3.81 per score increase, 95% CI, 1.01-14.36, P = 0.0479). (Please see Supplemental Table IV, results section)

Table IV. Test of Newly Developed Model for Independent set of Patients (N = 37)

Characteristic

LOS ≤ 14 days

(N = 23)

LOS > 14 days
(N= 14)

OR (95% CI)

P value

AUC
(95% CI)

Model 3

 Score (per score)

1.54  (0.97-2.45)

0.0686

0.655 (0.490-0.820)

  Score ≥ 0

17/23 (73.9%)

6/14 (42.9%)

  Score ≥ 1

0/23 (0%)

0/14 (0%)

  Score ≥ 2

0/23 (0%)

1/14 (7.1%)

  Score ≥ 3

6/23 (26.1%)

6/14 (42.9%)

  Score ≥ 4

0/23 (0%)

1/14 (7.1%)

  Score ≥ 5

0/23 (0%)

0/14 (0%)

  Score ≥ 6

0/23 (0%)

0/14 (0%)

Model 4

  Score (per score)

-

-

3.81  (1.01-14.36)

0.0479*

0.693 (0.565-0.812)

  Score ≥ 0

17/23 (73.9%)

6/14 (42.9%)

  Score ≥ 1

6/23 (26.1%)

7/14 (50.0%)

-

-

-

  Score ≥ 2

0/23 (0%)

1/14 (7.4%)

-

-

-

  Score ≥ 3

0/23 (0%)

0

-

-

-

Abbreviations: AUC, area under curve; ICU, intensive care unit; OR, odds ratio.

*Statistical significance (P < 0.05).

MINOR COMMENTS

  • The manuscript could be further improved by citing a few recent research papers that discuss MDW as a useful biomarker for sepsis/clinical severity in COVID-19.
    Riva et al, (2021) Scientific Reports 11:12716
    https://doi.org/10.1038/s41598-021-92236-6
    Alsuwaidi et al. (2022) BMC Infectious Diseases 22:27
    https://doi.org/10.1186/s12879-021-07016-4
  • Thank you for the comment. In our revised manuscript, we have cited the two studies and other related literature as follows. MDW is strongly associated with COVID-19[10-12]and can be used to distinguish COVID-19 from other upper airway infections[13]. The flow cytometry-based studies[14,15] showed immunophenotypic changes of peripheral monocytes in patients with COVID-19. In addition, MDW is found to be positively associated with several inflammatory acute phase proteins, including CRP, ferritin, and fibrinogens.[11,16] A study[16] determined that MDW > 24 was associated with unfavorable outcome in COVID-19. (Please see introduction section and references)

Reference:

  1. Chatterjee, N.A.; Jensen, P.N.; Harris, A.W.; Nguyen, D.D.; Huang, H.D.; Cheng, R.K.; Savla, J.J.; Larsen, T.R.; Gomez, J.M.D.; Du-Fay-de-Lavallaz, J.M.; et al. Admission respiratory status predicts mortality in COVID-19. Influenza Other Respir Viruses 2021, 15, 569-572, doi:10.1111/irv.12869.
  2. Berlin, D.A.; Gulick, R.M.; Martinez, F.J. Severe Covid-19. N Engl J Med 2020, 383, 2451-2460, doi:10.1056/NEJMcp2009575.
  3. Kazory, A.; Ronco, C.; McCullough, P.A. SARS-CoV-2 (COVID-19) and intravascular volume management strategies in the critically ill. Proc (Bayl Univ Med Cent) 2020, 0, 1-6, doi:10.1080/08998280.2020.1754700.
  4. Struyf, T.; Deeks, J.J.; Dinnes, J.; Takwoingi, Y.; Davenport, C.; Leeflang, M.M.; Spijker, R.; Hooft, L.; Emperador, D.; Domen, J.; et al. Signs and symptoms to determine if a patient presenting in primary care or hospital outpatient settings has COVID-19. Cochrane Database Syst Rev 2021, 2, CD013665, doi:10.1002/14651858.CD013665.pub2.
  5. Drummond, G.B.; Fischer, D.; Arvind, D.K. Current clinical methods of measurement of respiratory rate give imprecise values. ERJ Open Res 2020, 6, doi:10.1183/23120541.00023-2020.
  6. Kallioinen, N.; Hill, A.; Christofidis, M.J.; Horswill, M.S.; Watson, M.O. Quantitative systematic review: Sources of inaccuracy in manually measured adult respiratory rate data. J Adv Nurs 2021, 77, 98-124, doi:10.1111/jan.14584.
  7. Smith, G.B.; Prytherch, D.R.; Meredith, P.; Schmidt, P.E.; Featherstone, P.I. The ability of the National Early Warning Score (NEWS) to discriminate patients at risk of early cardiac arrest, unanticipated intensive care unit admission, and death. Resuscitation 2013, 84, 465-470, doi:10.1016/j.resuscitation.2012.12.016.
  8. Subbe, C.P.; Kruger, M.; Rutherford, P.; Gemmel, L. Validation of a modified Early Warning Score in medical admissions. QJM 2001, 94, 521-526, doi:10.1093/qjmed/94.10.521.
  9. Gardner-Thorpe, J.; Love, N.; Wrightson, J.; Walsh, S.; Keeling, N. The value of Modified Early Warning Score (MEWS) in surgical in-patients: a prospective observational study. Ann R Coll Surg Engl 2006, 88, 571-575, doi:10.1308/003588406X130615.
  10. Lippi, G.; Plebani, M. The critical role of laboratory medicine during coronavirus disease 2019 (COVID-19) and other viral outbreaks. Clinical Chemistry and Laboratory Medicine (CCLM) 2020, 58, 1063-1069.
  11. Riva, G.; Castellano, S.; Nasillo, V.; Ottomano, A.M.; Bergonzini, G.; Paolini, A.; Lusenti, B.; Milić, J.; De Biasi, S.; Gibellini, L. Monocyte Distribution Width (MDW) as novel inflammatory marker with prognostic significance in COVID-19 patients. Scientific Reports 2021, 11, 1-9.
  12. Alsuwaidi, L.; Al Heialy, S.; Shaikh, N.; Al Najjar, F.; Seliem, R.; Han, A.; Hachim, M. Monocyte distribution width as a novel sepsis indicator in COVID-19 patients. BMC infectious diseases 2022, 22, 1-10.
  13. Lin, H.-A.; Lin, S.-F.; Chang, H.-W.; Lee, Y.-J.; Chen, R.-J.; Hou, S.-K. Clinical impact of monocyte distribution width and neutrophil-to-lymphocyte ratio for distinguishing COVID-19 and influenza from other upper respiratory tract infections: A pilot study. Plos one 2020, 15, e0241262.
  14. Sánchez-Cerrillo, I.; Landete, P.; Aldave, B.; Sánchez-Alonso, S.; Sánchez-Azofra, A.; Marcos-Jiménez, A.; Ávalos, E.; Alcaraz-Serna, A.; de Los Santos, I.; Mateu-Albero, T.; et al. COVID-19 severity associates with pulmonary redistribution of CD1c+ DCs and inflammatory transitional and nonclassical monocytes. J Clin Invest 2020, 130, 6290-6300, doi:10.1172/JCI140335.
  15. Zhang, D.; Guo, R.; Lei, L.; Liu, H.; Wang, Y.; Qian, H.; Dai, T.; Zhang, T.; Lai, Y.; Wang, J.; et al. Frontline Science: COVID-19 infection induces readily detectable morphologic and inflammation-related phenotypic changes in peripheral blood monocytes. J Leukoc Biol 2021, 109, 13-22, doi:10.1002/JLB.4HI0720-470R.
  16. Alsuwaidi, L.; Al Heialy, S.; Shaikh, N.; Al Najjar, F.; Seliem, R.; Han, A.; Hachim, M. Monocyte distribution width as a novel sepsis indicator in COVID-19 patients. BMC Infect Dis 2022, 22, 27, doi:10.1186/s12879-021-07016-4.

Reviewer 2 Report

Line 125: LOS may be better defined, it is not clear if it refers to overall LOS in Hospital or to EDLOS (emergency department Los. 

Discussion may be improved,  for example from line 45, and you could specify how the nature of infection could affect MDW.

Author Response

We thank the reviewers for the constructive comments. We have made revisions to the manuscript to address all the questions and comments raised by the reviewers. We have highlighted changes made to the original version by setting the text color to red. Our specific responses to each comment are as follows:

Comments and Suggestions for Authors

  • Line 125: LOS may be better defined, it is not clear if it refers to overall LOS in Hospital or to EDLOS (emergency department Los).
  • Thank you for the comment. In our study, the length of stay (LOS) was calculated as the overall LOS in hospital rather than emergency department length of stay (EDLOS). In our hospital, patients with COVID-19 must be admitted to an isolation ward or the ICU within 6 h on arrival to ED. In our revised manuscript, we have more clearly defined the LOS as the time elapsed between a patient’s hospital admittance to isolation ward and discharge. (Please see 2.2 Data collection, Methods section)

  • Discussion may be improved, for example from line 45, and you could specify how the nature of infection could affect MDW.
  • Thank you for the comment. In our revised manuscript, we have mentioned monocytes are the key member of the mononuclear phagocyte system, a part of the innate immune system[1,2]. MDW is responsive to bacterial and viral pathogens[3-5], and the magnitude of increase of MDW is correlated with the intensity of infection and sepsis.[6-9] MDW is strongly associated with COVID-19[10-12]and can be used to distinguish COVID-19 from other upper airway infections[13]. The flow cytometry-based studies[14,15] showed immunophenotypic changes of peripheral monocytes in patients with COVID-19. In addition, MDW is found to be positively associated with several inflammatory acute phase proteins, including CRP, ferritin, and fibrinogens.[11,16] A study[16] determined MDW > 24 was associated with unfavorable outcome in COVID-19. (Please see 1. introduction section)

Reference:

  1. Henderson, R.B.; Hobbs, J.A.; Mathies, M.; Hogg, N. Rapid recruitment of inflammatory monocytes is independent of neutrophil migration. Blood 2003, 102, 328-335.
  2. Italiani, P.; Boraschi, D. From monocytes to M1/M2 macrophages: phenotypical vs. functional differentiation. Frontiers in immunology 2014, 514.
  3. Xu, D. Clinical applications of leukocyte morphological parameters. Int J Pathol Clin Res 2015, 1.
  4. McCullough, K.; Basta, S.; Knötig, S.; Gerber, H.; Schaffner, R.; Kim, Y.; Saalmüller, A.; Summerfield, A. Intermediate stages in monocyte–macrophage differentiation modulate phenotype and susceptibility to virus infection. Immunology 1999, 98, 203.
  5. Wang, S.; Mak, K.; Chen, L.; Chou, M.; Ho, C. Heterogeneity of human blood monocyte: two subpopulations with different sizes, phenotypes and functions. Immunology 1992, 77, 298.
  6. Crouser, E.D.; Parrillo, J.E.; Seymour, C.W.; Angus, D.C.; Bicking, K.; Esguerra, V.G.; Peck-Palmer, O.M.; Magari, R.T.; Julian, M.W.; Kleven, J.M. Monocyte distribution width: a novel indicator of sepsis-2 and sepsis-3 in high-risk emergency department patients. Critical care medicine 2019, 47, 1018.
  7. Crouser, E.D.; Parrillo, J.E.; Seymour, C.; Angus, D.C.; Bicking, K.; Tejidor, L.; Magari, R.; Careaga, D.; Williams, J.; Closser, D.R. Improved early detection of sepsis in the ED with a novel monocyte distribution width biomarker. Chest 2017, 152, 518-526.
  8. Piva, E.; Zuin, J.; Pelloso, M.; Tosato, F.; Fogar, P.; Plebani, M. Monocyte distribution width (MDW) parameter as a sepsis indicator in intensive care units. Clinical Chemistry and Laboratory Medicine (CCLM) 2021.
  9. Hou, S.-K.; Lin, H.-A.; Chen, S.-C.; Lin, C.-F.; Lin, S.-F. Monocyte distribution width, neutrophil-to-lymphocyte ratio, and platelet-to-lymphocyte ratio improves early prediction for sepsis at the emergency. Journal of personalized medicine 2021, 11, 732.
  10. Lippi, G.; Plebani, M. The critical role of laboratory medicine during coronavirus disease 2019 (COVID-19) and other viral outbreaks. Clinical Chemistry and Laboratory Medicine (CCLM) 2020, 58, 1063-1069.
  11. Riva, G.; Castellano, S.; Nasillo, V.; Ottomano, A.M.; Bergonzini, G.; Paolini, A.; Lusenti, B.; Milić, J.; De Biasi, S.; Gibellini, L. Monocyte Distribution Width (MDW) as novel inflammatory marker with prognostic significance in COVID-19 patients. Scientific Reports 2021, 11, 1-9.
  12. Alsuwaidi, L.; Al Heialy, S.; Shaikh, N.; Al Najjar, F.; Seliem, R.; Han, A.; Hachim, M. Monocyte distribution width as a novel sepsis indicator in COVID-19 patients. BMC infectious diseases 2022, 22, 1-10.
  13. Lin, H.-A.; Lin, S.-F.; Chang, H.-W.; Lee, Y.-J.; Chen, R.-J.; Hou, S.-K. Clinical impact of monocyte distribution width and neutrophil-to-lymphocyte ratio for distinguishing COVID-19 and influenza from other upper respiratory tract infections: A pilot study. Plos one 2020, 15, e0241262.
  14. Sánchez-Cerrillo, I.; Landete, P.; Aldave, B.; Sánchez-Alonso, S.; Sánchez-Azofra, A.; Marcos-Jiménez, A.; Ávalos, E.; Alcaraz-Serna, A.; de Los Santos, I.; Mateu-Albero, T.; et al. COVID-19 severity associates with pulmonary redistribution of CD1c+ DCs and inflammatory transitional and nonclassical monocytes. J Clin Invest 2020, 130, 6290-6300, doi:10.1172/JCI140335.
  15. Zhang, D.; Guo, R.; Lei, L.; Liu, H.; Wang, Y.; Qian, H.; Dai, T.; Zhang, T.; Lai, Y.; Wang, J.; et al. Frontline Science: COVID-19 infection induces readily detectable morphologic and inflammation-related phenotypic changes in peripheral blood monocytes. J Leukoc Biol 2021, 109, 13-22, doi:10.1002/JLB.4HI0720-470R.
  16. Alsuwaidi, L.; Al Heialy, S.; Shaikh, N.; Al Najjar, F.; Seliem, R.; Han, A.; Hachim, M. Monocyte distribution width as a novel sepsis indicator in COVID-19 patients. BMC Infect Dis 2022, 22, 27, doi:10.1186/s12879-021-07016-4.

Reviewer 3 Report

Dear Authors,

I read with interest your article entitled “Monocyte Distribution Width Predicts Length of Stay for Patients with COVID-19: A Pioneer Study”. I have some issues that I want to expose to you, to improve the quality of the article.

Introduction: I suggest writing again this paragraph, giving more attention to the aim of your study. The introduction deals with the general aspect of current pandemia, while little data are provided about the role of risk factors or markers related to LOS >14 days.

.Materials and Methods: Despite this section resulted clear and well-written, I reported some issues.

You could improve your study by adding a score (i.e. Charlson comorbidity Index) to stratify your population for the comorbidities and age, so you could verify if the validity of your markers were influenced by a pre-existing condition or not.

 SIRS is no more present in the SEPSI guideline, so I invite you to consider using another score as the guideline suggests.

You divided your population based on LOS, and patients were discharged after two consecutive negative results of PCR tests performed with an interval of at least 24 hours. In my opinion, this criterion could be a bias. Waiting for the results of the second swab your LOS could be longer. Furthermore, you did not provide data about LOS in both groups, please provide these details in text or a table

Statistical Analysis

For a proper data presentation, you should test data distribution (Shapiro Wilk test) and present the data according to the results.

The last suggestion is to evaluate to choose another title for your article. MDW is not the only parameter that you use for your score!

Author Response

We thank the reviewers for the constructive comments. We have made revisions to the manuscript to address all the questions and comments raised by the reviewers. We have highlighted changes made to the original version by setting the text color to red in the revised manuscript. Our specific responses to each comment are as follows:

I read with interest your article entitled “Monocyte Distribution Width Predicts Length of Stay for Patients with COVID-19: A Pioneer Study”. I have some issues that I want to expose to you, to improve the quality of the article.

  • Introduction: I suggest writing again this paragraph, giving more attention to the aim of your study. The introduction deals with the general aspect of current pandemia, while little data are provided about the role of risk factors or markers related to LOS >14 days.
  • Thank you for the comment. We have written the paragraphs to describe the risk factors and markers are associated with prolonged LOSs of COVID-19 patients as follows.
  • Recent studies indicated the older adults[1-3] and patients with multiple comorbidities[2,4-7] are more vulnerable and associated with prolonged LOSs for COVID-19. Furthermore, on evaluating the risk factors of COVID-19 patients, a recent artificial intelligence based model[8] found that respiratory failure with ventilator support was the most critical variable to influence the LOS (45.4 vs. 7.4 days) and the mortality in hospitals. In this model[8], older age, impaired kidney function, higher body mass index are linked to more severe disease. Other studies reported diabetes[9] and infection with human immunodeficiency virus[10] are related to unfavorable outcome for patients with COVID-19. However, there are limited studies on appropriately predicting LOS by using data collected in emergency department (ED) setting[11].
  • In addition, numerous biomarkers are considered potential predictors of the severity and LOS for patients with COVID-19. Patients with high severity of COVID-19 presented with higher proportions of increased white blood cell (WBC) counts (4% vs 4.8%) compared to mild to moderate patients.[12] Moreover, patients with more advanced infection exhibited increased counts of neutrophils and decreased counts of lymphocytes.[13] Several studies [8,12,14,15] showed thrombocytopenia was related to with severe infection and poor prognosis. A study[16] showed C-reactive protein (CRP) was useful for monitoring the progression of infection and to detect severe cases of COVID-19 in the early phase.[17] Procalcitonin(PCT) can reflect disease severity and co-infection in patients with COVID-19.[16,18] Other inflammatory markers, including neutrophil-to-lymphocyte ratio (NLR)[19,20] and platelet-to-lymphocyte ratio(PLR)[20,21], were associated with the severity of COVID-19 and potentially could be employed to predict LOS.

  • Materials and Methods: Despite this section resulted clear and well-written, I reported some issues. You could improve your study by adding a score (i.e. Charlson comorbidity Index) to stratify your population for the comorbidities and age, so you could verify if the validity of your markers were influenced by a pre-existing condition or not.
  • Thank you for the comment. In our revised manuscript, we have revised our analysis with the Charlson Comorbidity Index (CCI). Between the groups of length of stay (LOS) ≤ 14 and > 14 days, the CCI showed no significant difference (9 ± 2.0 vs. 2.7 ± 2.0, P = 0.8724). In the univariate analysis, the CCI was weakly associated with LOS > 14 days (OR: 1.22, 95% CI, 1.01-1.46, P = 0.0399). In the multivariable analysis, the CCI was found not associated with LOS > 14 days (OR: 0.95, 95% CI, 0.70-1.30, P = 0.7545). The analysis with and without adjustment of CCI were shown as follows (Supplemental Table II). The magnitude of the associations for fever (body temperature > 38°C), tachypnea (respiratory rate > 20/min), and MDW ≥ 21 were not changed.

  • Supplemental Table II. Multivariable models with and without adjustment of the Charlson Comorbidity Index for predicting length of stay >14 days (N = 120).

Models

With adjustment on CCI

Without adjustment on CCI

Characteristic

Model 1a

OR (95% CI)

P value

Model 1a

OR (95% CI)

P value

Variable

  Age > 60 years

1.85 (0.55–6.19)

0.3208

1.63 (0.64–4.14)

0.3018

  Sex (male vs. female)

1.48 (0.60–3.67)

0.4001

1.45 (0.59–3.57)

0.4188

  BT > 38°C

2.46 (0.92–6.55)

0.0717

2.52 (0.96–6.63)

0.0609

  RR > 20/min

3.74 (1.12–12.54)

0.0320*

3.60 (1.11–11.64)

0.0329*

  SpO2 < 96%

0.78 (0.28–2.20)

0.6356

0.81 (0.29–2.22)

0.6764

  Hypertension

1.55 (0.60–4.01)

0.3720

1.48 (0.59–3.72)

0.4002

  MDW ≥ 21

4.72 (0.92–24.15)

0.0624

4.66 (0.92–23.75)

0.0639

  CRP > 3 mg/dL

0.88 (0.29–2.69)

0.8274

0.86 (0.29–2.56)

0.7824

  NLR < 3

1.68 (0.54–5.22)

0.3705

1.74 (0.57–5.30)

0.3287

  CCI

0.95 (0.70–1.30)

0.7545

-

-

Model fit

  AUC (95% CI)

0.787 (0.701–0.874)

0.786 (0.700–0.872)

  AIC

161.68

161.68

  Hosmer–Lemeshow test

8.785 (10 groups)

0.3607

9.106 (10 groups)

0.3334

Abbreviations: AUC, area under curve; BT, body temperature; CCI, CRP, C-reactive protein; NLR, neutrophil-to-lymphocyte ratio; OR, odds ratio; RR, respiratory rate; AIC, Akaike information criterion.

aModel 1 included all significant predictors from the multivariable analysis.

*Statistical significance (P < 0.05).

  • SIRS is no more present in the SEPSI guideline, so I invite you to consider using another score as the guideline suggests.
  • Thank you for the comment. The currently recommended approach to sepsis is based on two types of definitions: (1) in the Sepsis-2 criteria, sepsis is defined as presence of infection in conjugation with a systemic inflammatory response syndrome (SIRS) score of ≥ 2 points; and (2) in the new Sepsis-3 criteria, sepsis is defined as a quick sequential organ failure assessment (qSOFA) score of ≥ 2 followed by an acute increase of ≥2 points in the SOFA score due to the infection. The qSOFA score was composed by three items: altered mental status (Glasgow coma scale < 15), respiratory rate ≥ 22 /min, and systolic blood pressure ≤ 100 mmHg. In our revised manuscript, we have added the analysis with qSOFA score. Since our model was based on the data in the emergency department (ED), we assessed the sepsis with SIRS and qSOFA scores. The mean qSOFA were mildly increased in the group of length of stay (LOS) compared to the group of LOS ≤ 14 days (0.1 ± 0.3 vs. 0.5 ± 0.7, P = 0.0011). In the qSOFA, the group of LOS > 14 day had higher qSOFA score due to the parameter of respiratory rate ≥ 22 breaths /min (8.3% vs. 35.4%, P = 0.0002).
  • Our analysis should be compatible with an earlier study[22] which used the qSOFA to assess the critically ill patients with COVID-19. In this study[22], all 52 critical ill patients were analyzed. Of them, the mean qSOFA score for patiens with and without mechanical ventilation were both < 1 and showed no significant difference[22]. These patients with qSOFA ≥ 1 scored for the item respiratory rate ≥ 22/min. Our study reproduced the results (Table 1). For these patients, the qSOFA score can be linearly predicted form respiratory rate. Under this circumference, we did not include the qSOFA score due to the high collinearity between qSOFA and respiratory rate in the logistic regression analysis.

  • Table 1. Patient characteristics (N = 120).

Characteristic

LOS ≤ 14 days
(N = 72)

LOS > 14 days

(N = 48)

P value

Vital signs at ED

    Body temperature (°C)†

37.0 ± 0.8

37.4 ± 1.0

0.0277*

    Heart rate (beats/min)

88.7 ± 15.9

91.8 ± 18.7

0.3298

    Respiratory rate (breaths/min) †

18.3 ± 2.6

21.3 ± 4.2

<0.0001*

    SBP (mmHg)

131.1 ± 19.2

132.0 ± 22.9

0.8132

    DBP (mmHg)

80.1 ± 14.7

78.5 ± 14.7

0.5658

    MAP (mmHg)†

97.1 ± 14.6

96.3 ± 15.4

0.9340

    SIRS score†

1.4 ± 0.9

1.6 ± 0.9

0.1872

    qSOFA score†

0.1 ± 0.3

0.5 ± 0.7

0.0011*

      Glasgow coma scale < 15

2/72 (2.8%)

3/48 (6.3%)

0.2320

      Respiratory rate ≥ 22 /min

6/72 (8.3%)

17/48 (35.4%)

0.0002*

      SBP≤100 mmHg

2/72 (2.8%)

4/48 (8.3%)

0.1733

LOS, length of stay; DBP, diastolic blood pressure; MAP, mean arterial pressure; qSOFA, the quick Sequential Organ Failure Assessment; RDW, red distribution width; SBP, systolic blood pressure; SIRS, systemic inflammatory response syndrome;  

*Statistical significance (P < 0.05).

†The Mann-Whitney U test was used.

  • You divided your population based on LOS, and patients were discharged after two consecutive negative results of PCR tests performed with an interval of at least 24 hours. In my opinion, this criterion could be a bias. Waiting for the results of the second swab your LOS could be longer. Furthermore, you did not provide data about LOS in both groups, please provide these details in text or a table
  • Thank you for the comment. In our study, all patients with COVID-19 were discharged when their acute medical problems were treated, and all enrolled patients completed the second swab within 24-48 h. In other words, our study may have a LOS bias of one day. These patients may have LOS bias of one day due to the measurement time of the second swab. We have addressed that the waiting time for the results of the second swab could be bias in our study. In our revised manuscript, we have provided the mean LOS (7 ± 1.9 versus 28.4 ± 16.8 days) and median LOS (11 and 21 days) in Table 1. We have compared the LOS in both groups with Mann-Whitney U test (P < 0.0001).

  • Statistical Analysis
    For a proper data presentation, you should test data distribution (Shapiro Wilk test) and present the data according to the results.
  • Thank you for the comment. In our revised manuscript, we performed the Shapiro-Wilk test to assess the normality of the variables (Please see Supplemental Table I). Accordingly, the characteristics between the groups of length of stay (LOS) ≤ 14 and > 14 days were analyzed using the Student’s t test if variables fulfilled normal distribution, the Mann-Whitney U test if variables violated normal distribution. Accordingly, we have revised the Table 1 by using the Mann-Whitney U test. On the other hands, the logistic regression model does not require a linear relationship between the dependent and independent variables, the residuals do not require to be normally distributed, and no assumption of homoscedasticity is required. Our logistic regression analysis was still valid.

Supplemental Table I. Assessment of continuous variable distribution (N = 120)

Variables

Minimum

Maximum

Mean ± SD

Shapiro–Wilk test

Statistics

P value

Age

17

91

56.5 ± 17.0

0.9775

0.0417*

BMI (kg/m2)

17.6

42.5

25.2 ± 5.0

0.9219

<0.0001*

Body temperature (°C)

35.0

39.2

37.2 ± 0.9

0.9511

<0.0001*

Heart rate (beats/min)

48

134

90.0 ± 17.1

0.9833

0.1451

Respiratory rate (breaths/min)

12

32

19.5 ± 3.6

0.8257

<0.0001*

SpO2 (%)

63

100

95.3 ± 5.1

0.7404

<0.0001*

SBP (mmHg)

78

197

131.5 ± 20.7

0.9795

0.0633

DBP (mmHg)

51

128

79.4 ± 14.7

0.9809

0.0870

MAP (mmHg)

60

151

96.8 ± 14.9

0.9773

0.0401*

SIRS score

0

4

1.5 ± 0.9

0.8793

<0.0001*

LOS (days)

4

90

17.8 ± 13.8

0.6605

<0.0001*

WBC (103 cells/μL)

2.6

19.9

6.5 ± 2.7

0.8787

<0.0001*

CRP (mg/dL)

0.1

31.8

5.2 ± 5.7

0.8299

<0.0001*

PCT (ng/mL)

0.03

7.89

0.33 ± 1.01

0.3058

<0.0001*

MDW

15.9

42.2

24.7 ± 4.5

0.9616

<0.0001*

RDW (%)

12.1

21.1

13.6 ± 1.2

0.7364

<0.0001*

NLR

0.9

42.5

6.0 ± 6.9

0.6593

<0.0001*

PLR

39.7

1144.5

216.0 ± 166.9

0.7031

<0.0001*

Charlson Comorbidity Index

0

8

2.2 ± 2.0

0.8930

<0.0001*

Ct number

12

36

22.7 ± 6.0

0.9536

0.0006*

LOS, length of stay; CRP, C-reactive protein; Ct, cycle threshold value for COVID-19 positive cases; DBP, diastolic blood pressure; ED, emergency department; ICU, intensive care unit; MAP, mean arterial pressure; MDW, monocyte distribution width; NLR, neutrophil-to-lymphocyte ratio; PCT, procalcitonin; PLR, platelet-to-lymphocyte ratio; SBP, systolic blood pressure; SD, standard deviation; SIRS, systemic inflammatory response syndrome; SpO2, peripheral capillary oxygen saturation.

*Statistical significance (P < 0.05).

  • The last suggestion is to evaluate to choose another title for your article. MDW is not the only parameter that you use for your score!
  • Thank you for the comment. In our newly developed scoring model, we found fever > 38°C, tachypnea > 20 breath/min, and MDW ≥21 were significant predictors for increased length of stay. In our revised manuscript, we have replaced the original title with “Fever, Tachypnea, and Monocyte Distribution Width Predicts Length of Stay for Patients with COVID-19: A Pioneer Study.” (Please see Title, page 1)

Reference:

  1. Yanez, N.D.; Weiss, N.S.; Romand, J.-A.; Treggiari, M.M. COVID-19 mortality risk for older men and women. BMC Public Health 2020, 20, 1-7.
  2. Clark, A.; Jit, M.; Warren-Gash, C.; Guthrie, B.; Wang, H.H.; Mercer, S.W.; Sanderson, C.; McKee, M.; Troeger, C.; Ong, K.I. How many are at increased risk of severe COVID-19 disease? Rapid global, regional and national estimates for 2020. MedRxiv 2020.
  3. Lewnard, J.A.; Liu, V.X.; Jackson, M.L.; Schmidt, M.A.; Jewell, B.L.; Flores, J.P.; Jentz, C.; Northrup, G.R.; Mahmud, A.; Reingold, A.L. Incidence, clinical outcomes, and transmission dynamics of hospitalized 2019 coronavirus disease among 9,596,321 individuals residing in California and Washington, United States: a prospective cohort study. MedRxiv 2020.
  4. Di Castelnuovo, A.; Bonaccio, M.; Costanzo, S.; Gialluisi, A.; Antinori, A.; Berselli, N.; Blandi, L.; Bruno, R.; Cauda, R.; Guaraldi, G. Common cardiovascular risk factors and in-hospital mortality in 3,894 patients with COVID-19: survival analysis and machine learning-based findings from the multicentre Italian CORIST Study. Nutrition, Metabolism and Cardiovascular Diseases 2020, 30, 1899-1913.
  5. Mehmood, I.; Ijaz, M.; Ahmad, S.; Ahmed, T.; Bari, A.; Abro, A.; Allemailem, K.S.; Almatroudi, A.; Tahir ul Qamar, M. SARS-CoV-2: An update on genomics, risk assessment, potential therapeutics and vaccine development. International Journal of Environmental Research and Public Health 2021, 18, 1626.
  6. Rodriguez-Morales, A.J.; Cardona-Ospina, J.A.; Gutiérrez-Ocampo, E.; Villamizar-Peña, R.; Holguin-Rivera, Y.; Escalera-Antezana, J.P.; Alvarado-Arnez, L.E.; Bonilla-Aldana, D.K.; Franco-Paredes, C.; Henao-Martinez, A.F. Clinical, laboratory and imaging features of COVID-19: A systematic review and meta-analysis. Travel medicine and infectious disease 2020, 34, 101623.
  7. Yang, J.; Zheng, Y.; Gou, X.; Pu, K.; Chen, Z.; Guo, Q.; Ji, R.; Wang, H.; Wang, Y.; Zhou, Y. Prevalence of comorbidities and its effects in patients infected with SARS-CoV-2: a systematic review and meta-analysis. International Journal of Infectious Diseases 2020, 94, 91-95.
  8. Mahboub, B.; Al Bataineh, M.T.; Alshraideh, H.; Hamoudi, R.; Salameh, L.; Shamayleh, A. Prediction of COVID-19 hospital length of stay and risk of death using artificial intelligence-based modeling. Frontiers in medicine 2021, 8.
  9. Mota, M.; Stefan, A.-G. Covid-19 and Diabetes–A Bidirectional Relationship? Romanian Journal of Diabetes Nutrition and Metabolic Diseases 2020, 27, 77-79.
  10. Ssentongo, P.; Heilbrunn, E.S.; Ssentongo, A.E.; Advani, S.; Chinchilli, V.M.; Nunez, J.J.; Du, P. Epidemiology and outcomes of COVID-19 in HIV-infected individuals: a systematic review and meta-analysis. Scientific reports 2021, 11, 1-12.
  11. Rees, E.M.; Nightingale, E.S.; Jafari, Y.; Waterlow, N.R.; Clifford, S.; Pearson, C.A.; Jombart, T.; Procter, S.R.; Knight, G.M.; Group, C.W. COVID-19 length of hospital stay: a systematic review and data synthesis. BMC medicine 2020, 18, 1-22.
  12. Lippi, G.; Plebani, M. The critical role of laboratory medicine during coronavirus disease 2019 (COVID-19) and other viral outbreaks. Clinical Chemistry and Laboratory Medicine (CCLM) 2020, 58, 1063-1069.
  13. Fan, B.E. Hematologic parameters in patients with COVID-19 infection: a reply. American journal of hematology 2020.
  14. Liu, Y.; Sun, W.; Guo, Y.; Chen, L.; Zhang, L.; Zhao, S.; Long, D.; Yu, L. Association between platelet parameters and mortality in coronavirus disease 2019: Retrospective cohort study. Platelets 2020, 31, 490-496.
  15. Zulfiqar, A.-A.; Lorenzo-Villalba, N.; Hassler, P.; Andrès, E. Immune thrombocytopenic purpura in a patient with Covid-19. New England Journal of Medicine 2020, 382, e43.
  16. Henry, B.M.; Lippi, G.; Plebani, M. Laboratory abnormalities in children with novel coronavirus disease 2019. Clinical Chemistry and Laboratory Medicine (CCLM) 2020, 58, 1135-1138.
  17. Tan, C.; Huang, Y.; Shi, F.; Tan, K.; Ma, Q.; Chen, Y.; Jiang, X.; Li, X. C‐reactive protein correlates with computed tomographic findings and predicts severe COVID‐19 early. Journal of medical virology 2020, 92, 856-862.
  18. Lippi, G.; Plebani, M. Laboratory abnormalities in patients with COVID-2019 infection. Clinical Chemistry and Laboratory Medicine (CCLM) 2020, 58, 1131-1134.
  19. Qin, C.; Zhou, L.; Hu, Z.; Zhang, S.; Yang, S.; Tao, Y.; Xie, C.; Ma, K.; Shang, K.; Wang, W. Dysregulation of immune response in patients with coronavirus 2019 (COVID-19) in Wuhan, China. Clinical infectious diseases 2020, 71, 762-768.
  20. Peng, J.; Qi, D.; Yuan, G.; Deng, X.; Mei, Y.; Feng, L.; Wang, D. Diagnostic value of peripheral hematologic markers for coronavirus disease 2019 (COVID‐19): A multicenter, cross‐sectional study. Journal of clinical laboratory analysis 2020, 34, e23475.
  21. Sarkar, S.; Kannan, S.; Khanna, P.; Singh, A.K. Role of platelet‐to‐lymphocyte count ratio (PLR), as a prognostic indicator in COVID‐19: A systematic review and meta‐analysis. Journal of Medical Virology 2022, 94, 211-221.
  22. Ferreira, M.; Blin, T.; Collercandy, N.; Szychowiak, P.; Dequin, P.F.; Jouan, Y.; Guillon, A. Critically ill SARS-CoV-2-infected patients are not stratified as sepsis by the qSOFA. Ann Intensive Care 2020, 10, 43, doi:10.1186/s13613-020-00664-w.

Round 2

Reviewer 3 Report

Dear authors

I appreciated the changes you made, especially in the methods section. However, several points need further clarification.

Introduction

In this section, you added several details and literature references. However, it appeared too long and quite confusing. As a result, the reader could find reading and understanding your study challenging. Therefore, I suggest organising the introduction, especially to make it shorter.

Materials and methods

2.3 Clinical spectrum of Covid-19

I suggest presenting the COVID-19 classification (from lines 140 to 150) in a table or picture.

Results

Line 200: You reported: "Between the groups of LOSs of ≤14 and > 14 days, the mean LOSs were 10.7 ± 1.9 and 28.4 ± 16.8 days, and the median LOSs were 11 and 21 days." Was this data correct about the last value 28.4 ±16.8 days for LOS>14 days? Is the SD correct? Please verify this aspect.

However, the results section was confusingly and redundant. I think that you should not report data already described in the tables. Furthermore, in table 2, why did you write two cutoff values (20,21) for MDW when in the same table, you established a cutoff of 21? Why, for NLR did you choose three as the cutoff?

The results presentations should be improved, and I suggest you control the data presented in the tables.

Discussion

 I suggest organising the discussion, especially to make it shorter.

In the end, you add a lot of information, but without modifying the original work, and It now results in too long and confusing.

Author Response

Dear authors
I appreciated the changes you made, especially in the methods section. However, several points need further clarification.

  • We are grateful for the constructive comments. We have made revisions to the manuscript to address all the questions and comments and have highlighted changes made to the original version by setting the text color to red in the revised manuscript. Our specific responses to each comment are as follows:

Introduction
In this section, you added several details and literature references. However, it appeared too long and quite confusing. As a result, the reader could find reading and understanding your study challenging. Therefore, I suggest organising the introduction, especially to make it shorter.

  • Thank you for the comment. We have deleted the redundant text describing the pandemic of COVID-19 and have rewritten the introduction to make it shorter. In our revised manuscript, there were only four paragraphs in introduction section: (1) the first paragraph described the impact of COVID-19; (2) the second paragraph reviewed the factors associated with length of stay (LOS) of COVID-19; (3) the third paragraph was a literature regarding inflammatory biomarkers; (4) the last paragraph highlighted how the new biomarker monocyte distribution width (MDW) could be used to predict LOS for patients with COVID-19.
  • (Please see introduction section)

Materials and methods
3 Clinical spectrum of Covid-19
I suggest presenting the COVID-19 classification (from lines 140 to 150) in a table or picture.

  • Thank you for the comment. In our revised manuscript, the clinical spectrum of COVID-19 was presented in supplemental Table 1 as follows.

Supplemental Table I. Clinical Spectrum of Coronavirus disease 2019 (COVID-19) Infection

Clinical Spectrum

Clinical Presentation

Dyspnea

Chest x-ray

SpO2

Shock

Mild illness

Patients had any symptoms and signs of COVID-19 (including fever, cough, sore throat, diarrhea, and loss of taste or smell)  but did not exhibit shortness of breath, dyspnea

No

Normal

≥94% at room air

No

Moderate illness

Patients had any symptoms and signs of COVID-19 and lower respiratory infection with an oxygen saturation (SpO2) of ≥94% at room air

Presence

Infiltrates ≤ 50%

≥94% at room air

No

Severe illness

Patients had any above symptoms and signs of COVID-19 with an SpO2 of <94% at room air, a ratio of arterial partial pressure of oxygen to fraction of inspired oxygen (PaO2/FiO2) of <300 mmHg, a respiratory rate of >30 breaths/min, or a chest X-ray with lung infiltrates >50%

Presence, or  a respiratory rate of >30 breaths/min

Infiltrates >50%

<94% at room air, or (PaO2/FiO2) of <300 mmHg

No

Critical illness

Patients with any above symptoms and signs and exhibited respiratory failure, septic shock, or multiple organ failures

Presence

Infiltrates >50%

<94%, or (PaO2/FiO2) of <300 mmHg

Presence

SpO2, peripheral capillary oxygen saturation

Results
Line 200: You reported: "Between the groups of LOSs of ≤14 and > 14 days, the mean LOSs were 10.7 ± 1.9 and 28.4 ± 16.8 days, and the median LOSs were 11 and 21 days." Was this data correct about the last value 28.4 ±16.8 days for LOS>14 days? Is the SD correct? Please verify this aspect.

  • Thank you for the comment. We have checked the distribution of length of stay (LOS) in our patients and verified the SD for LOS. Compared to the group of LOS of ≤14 days, the group of LOS > 14 days showed the wilder distribution for LOSs. While patients in the group of LOS of ≤14 days had the minimum LOS of 4 days and maximum LOS of 14 days, patients in the group of > 14 days had the minimum LOS of 14 days and maximum LOS of 90 days.

However, the results section was confusingly and redundant. I think that you should not report data already described in the tables. Furthermore, in table 2, why did you write two cutoff values (20,21) for MDW when in the same table, you established a cutoff of 21? Why, for NLR did you choose three as the cutoff?

  • Thank you for the comment. In our revised manuscript, we have simplified the description in the results section. We have controlled the length of Table 1-4, and have avoided reporting data already in Table 1-4. (Please see results section)
  • An earlier study demonstrated that monocyte distribution width (MDW) > 20 was associated with sepsis detection. We agreed that this was redundant and we deleted the superfluous part. (Table 2) In addition, the optimal cutoff of NLR was determined by the Youden’s index (the point with maximum value of sensitivity + specificity − 1). The receiver of the operating characteristic curve showed Youden’s index of NLR was 3.105. Therefore, we chose 3 as the cutoff value of NLR. In our revised manuscript, we added the footnote in Table 2 to highlight that the cutoff value was determined by Youden’s index.

The results presentations should be improved, and I suggest you control the data presented in the tables.

  • Thank you for the comment. We have controlled the length of tables in the main manuscript, and have avoided redundant reporting data already in Table 1-4. The additional data, including age group, detailed vital signs at emergency department, and medical comorbidities, were moved to supplemental Table II. The data regarding diagnostic performance of each predictor (including positive predictive value, negative predictive value, and accuracy) were moved to supplemental Table III.

Discussion

I suggest organising the discussion, especially to make it shorter.

  • Thank you for the comment. In our revised manuscript, we have organized the discussion section and shortened the length to one page. In the first three paragraphs, we highlighted the tachypnea, fever, and elevated MDW were key predictors of prolonged LOS associated with COVID-19. In the fourth paragraph, we explained why the developed Model 4 should be the favored model for clinical practice. In the fifth paragraph, we clarified why qSOFA score was not appropriate for applying to predict LOS of COVID-19. In the last paragraph, we summarized the limitations in our study.

In the end, you add a lot of information, but without modifying the original work, and It now results in too long and confusing.

  • Thank you for the comment. We have modified the original work and revised our text. Our manuscript was more succinct. (Please see our revised manuscript with highlighted changes made to the original version by setting the text color to red in the revised manuscript)
